# Learning Fairer Representations with FairVIC

## Abstract

Mitigating bias in automated decision-making systems, specifically deep learning models, is a critical challenge in achieving fairness. This complexity stems from factors such as nuanced definitions of fairness, unique biases in each dataset, and the trade-off between fairness and model accuracy. To address such issues, we introduce FairVIC, an innovative approach designed to enhance fairness in neural networks by addressing inherent biases at the training stage. Unlike other methods that require a user-defined declaration of what it means to be fair, FairVIC integrates an abstract concept of fairness through variance, invariance and covariance terms into the loss function. These terms aim to minimise the model's dependency on protected characteristics for making predictions, thus promoting fairness. Our experimentation consists of evaluating FairVIC against other comparable bias mitigation techniques, on a number of datasets known for their biases. Additionally, we conduct an ablation study to examine the accuracy-fairness trade-off. We also extend FairVIC by offering multi-objective lambda recommendations, allowing users to train a fairer model with a set of weights that are tuned best for their application. Through our implementation of FairVIC, we observed a significant improvement in fairness across all metrics tested, without compromising the model's accuracy. Our findings suggest that FairVIC presents a straightforward, out-of-the-box solution for the development of fairer deep learning models, thereby offering a generalisable solution applicable across many tasks and datasets.

## 1 Introduction

With the ever-increasing utilisation of Artificial Intelligence (AI) in everyday applications, Neural Networks (NNs) have emerged as pivotal tools for Automated Decision Making (ADM) systems in industries such as healthcare (Esteva et al., 2017), finance (Dixon et al., 2017), and recruitment (Vardarlier & Zafer, 2020). However, the inherent bias embedded in datasets and subsequently learned by these models poses significant challenges to fairness. Such a bias can lead to adverse decisions affecting real lives. For instance, several studies have shown how bias in facial recognition technologies disproportionately misidentifies individuals of certain ethnic backgrounds (Birhane, 2022; Cavazos et al., 2020), leading to potential discrimination in law enforcement and hiring practices.

Real-world consequences exemplify the urgent need to address these challenges at the core of AI development. Ensuring fairness in deep learning models presents complex challenges, primarily due to the black-box nature of these models, which often complicates understanding and interpreting decisions. Moreover, the dynamic and high-dimensional nature of the data involved, combined with nuances in fairness definitions, further complicates the detection and correction of bias. This complexity necessitates the development of more sophisticated, inherently fair algorithms.

Previous mitigation strategies dealing with algorithmic bias – whether through pre-processing, in-processing, or post-processing – have significant limitations. Pre-processing techniques, which attempt to cleanse biased data, are labour-intensive, dependent on expert intervention (Salimi et al., 2019), and often insufficient to eliminate all biases. Current in-processing methods frequently lead to unstable models and often rely heavily on ad-hoc fairness metrics. Post-processing techniques, which adjust model predictions directly, ignore deeper issues without addressing the underlying biases in the data and model. These approaches lack stability, generalisability, and the ability to ensure fairness across multiple metrics (Berk et al., 2017).

In this paper, we introduce FairVIC (**Fair**ness through **V**ariance, **I**nvariance, and **C**ovariance), a novel approach that embeds fairness directly into neural networks by optimising a custom loss function. This function is designed to minimise the correlation between decisions and protected characteristics while maximising overall prediction performance. FairVIC integrates fairness through the concepts of variance, invariance, and covariance during the training process, making it more principled and intuitive, and universally applicable to diverse datasets. Unlike previous methods, FairVIC offers a robust, out-of-the-box solution that introduces a more abstract concept of fairness to significantly reduce bias. Our experimental evaluations demonstrate FairVIC's ability to significantly improve performance in all fairness metrics tested without compromising prediction accuracy. We compare our proposed method against comparable in-processing bias mitigation techniques, such as adversarial and constraint approaches, and highlight the improved, robust performance of the FairVIC model.

Our contributions in this paper are multi-fold:

- A novel, out-of-the-box in-processing bias mitigation technique for neural networks.
- An extension to dynamically tune the lambda weights during training, alleviating hyperparameter selection and fine-tuning from the user.
- A comprehensive experimental evaluation, using a multitude of comparable methods on a variety of metrics across several datasets, including different modalities such as tabular and text.
- An extended analysis of our proposed method to examine its robustness.

This paper is structured as follows: Section 2 discusses current approaches to mitigating bias throughout each processing stage. Section 3 describes any preliminary details for this work, including the fairness metrics used in the evaluation. Section 4 outlines our method and its lambda-tuning extension, including how each term in our loss function is calculated and an algorithm detailing how these terms are applied. Section 5 describes the experiments carried out, Section 6 outlines the results with discussion, and Section 7 concludes this work. Any extra information, including the dataset metadata and any extra experiments, is to be found in the Appendix.

## 2    RELATED WORK

There exist three broad categories of mitigation strategies for algorithmic bias: pre-processing, in-processing, and post-processing. Each aims to increase fairness differently by acting upon either the training data, the training step, or the predictions outputted by the model, respectively.

**Pre-processing** methods aim to *fix* the data before training, recognising that bias is primarily an issue with the data itself (Caton & Haas, 2020). In practice, this can be done a number of different ways, such as representative sampling, or re-sampling the data to reflect the full population (Shekhar et al., 2021; Ustun et al., 2019), reweighing the data such that different groups influence the model in a representative way (Calders & Žliobaitė, 2013; Kamiran & Calders, 2012), or generating synthetic data to balance out the representation of each group (Jang et al., 2021). Another set of approaches utilises causal methods to delineate relationships between sensitive attributes and the target variables within the data (Chiappa & Isaac, 2019; Kusner et al., 2017; Russell et al., 2017). Such techniques as these are labour-intensive and do not generalise well, requiring an expert with knowledge of the data to manually process each case of a new dataset (Salimi et al., 2019). They also cannot provide assurances that all bias has been removed – a model may draw upon relationships between features that lead to bias, which are hard for the expert to spot.

**In-processing** methods aim to train models to make fairer predictions, even upon biased data. There are a plethora of ways in which this has been done. For example, Celis et al. (2019) and Agarwal et al. (2018) utilise a chosen fairness metric and perform constraint optimisation during training. This has the effect that a single fairness metric needs to be chosen, introducing human bias, and this metric must perfectly capture the bias within the data to effectively mitigate it. Therefore, fairness cannot be achieved across multiple definitions in this way. Another approach involves incorporating an adversarial component during model training that penalises the model if protected characteristics can be predicted from its outputs (Zhang et al., 2018; Wadsworth et al., 2018; Xu et al., 2019). These methods are often effective but their main shortcoming is seen in their instability. Finally, the most relevant comparisons from previous work to our proposed method are regularisation-based

techniques that incorporate fairness constraints or penalties directly into the model's loss function during training. There are a number of ways that this can be done, such as through data augmentation strategies to promote less sensitive decision boundaries (Chuang & Mroueh, 2021) or by incorporating fairness adjustments into the boosting process (Cruz et al., 2023). The performance of these models differs from approach to approach, and those that work by constraining the model by a fairness metric directly suffer from the issue of human bias and misrepresenting the bias within the data/model.

**Post-processing** techniques involve adjusting model predictions or decision rules after training to ensure fair outcomes. In practice, decision thresholds have been adjusted for different groups to achieve equal outcomes in a particular metric (Hardt et al., 2016). Alternatively, labels near the decision boundary can be altered to favour less biased outcomes (Kamiran et al., 2012; 2018). Calibration (Kim et al., 2018; Noriega-Campero et al., 2019) adjusts the predictions of the model directly so that the proportion of positive instances is equal across each sub-group. This category of methods can oversimplify fairness, and they do not fix the underlying issue within the model. For those techniques that require the specification of a single fairness metric, the same issue applies surrounding this choice as before.

There lies a number of issues which have not yet been solved in parallel within one technique. These are: stability, generalisability, equal improvements to fairness across metrics (Berk et al., 2017), and built without requirements for user-induced definitions of fairness. In this paper, we solve all these requirements for an effective, out-of-the-box approach to mitigate bias through FairVIC.

## 3 PRELIMINARIES

### 3.1 VICREG

Variance-Invariance-Covariance Regularization (VICReg) (Bardes et al., 2021) has previously been used in self-supervised learning to tackle feature collapse and redundancy. It maximises variance across features to ensure the model produces diverse outputs for different inputs, minimises invariance between augmented representations of the same input to enhance stability, and reduces covariance among features to capture a broader range of information. VICReg is confined to this specific context and objective, and the application of these principles outside of self-supervised methods remains largely unexplored. In contrast, FairVIC extends these principles to supervised learning for bias mitigation. This adaptation addresses the challenges of fairness in decision-making systems, expanding the application of VIC principles beyond their original scope and offering a novel, generalisable solution to fairness in supervised learning models.

### 3.2 GROUP FAIRNESS METRICS

In this section, we introduce notation and state the fairness measures that we use to quantify bias.

**Equalized Odds Difference** requires that both the True Positive Rate (TPR) and False Positive Rate (FPR) are the same across groups defined by the protected attribute, where $TPR = \frac{TP}{TP+FN}$ and $FPR = \frac{FP}{FP+TN}$ (Hardt et al., 2016). Therefore, we calculate $\max(|FPR_p - FPR_u|, |TPR_p - TPR_u|)$, where $u$ represents the unprivileged groups and $p$ the privileged group and 0 signifies fairness.

**Average Absolute Odds Difference** averages the absolute differences in the false positive rates and true positive rates between groups, defined as $\frac{1}{2}(|FPR_u - FPR_p| + |TPR_u - TPR_p|)$, where $u$ represents the unprivileged groups and $p$ the privileged group, with 0 signifying fairness.

**Demographic Parity Difference** evaluates the difference in the probability of a positive prediction between groups, aiming for 0 to signify fairness. Formally, $DP = |P(\hat{Y} = 1|u) - P(\hat{Y} = 1|p)|$, where $u$ represents the unprivileged groups, $p$ the privileged group, and $\hat{Y} = 1$ a positive prediction (Dwork et al., 2012).

**Disparate Impact** compares the proportion of positive outcomes for the unprivileged group to that of the privileged group, with a ratio of 1 indicating no disparate impact, and therefore fairness.

Denoted as $DI = \frac{P(\hat{Y}=1|u)}{P(\hat{Y}=1|p)}$, where $u$ represents the unprivileged groups, $p$ the privileged group, and $\hat{Y} = 1$ a positive prediction (Feldman et al., 2015).

### 3.3 INDIVIDUAL FAIRNESS

While FairVIC aims to increase group fairness, the invariance term promotes direct improvements in individual fairness. This can be observed in our evaluations through counterfactual fairness. Counterfactual fairness ensures that decisions made by an algorithm are fair even when considering hypothetical (counterfactual) scenarios. For each individual, the sensitive attribute is switched to assess the model's ability to perform equally in both the original and counterfactual scenarios.

Formally, if $u$ denotes the unprivileged group, $p$ the privileged group and $\hat{Y}$ is the decision outcome, then the model is considered counterfactually fair if $\hat{Y}_u = \hat{Y}_p$ for different groups $u$ and $p$ of the sensitive attribute while all non-sensitive features remain the same.

## 4 APPROACH

We propose FairVIC (**Fair**ness through **V**ariance, **I**nvariance, and **C**ovariance), a novel loss function that enables a model's ability to learn fairness in a robust manner. FairVIC is comprised of three terms: variance, invariance, and covariance. Optimising for these three terms encourages the model to be stable and consistent across protected characteristics, thereby reducing bias during training. By adopting this broad, generalised approach to defining bias, FairVIC significantly improves performance across a range of fairness metrics. This makes it an effective strategy for reducing bias across various applications, ensuring more equitable outcomes in diverse settings.

### 4.1 FAIRVIC TRAINING

To understand how FairVIC operates, it is crucial to define variance, invariance, and covariance within the context of fairness:

**Variance**: This aims to stop stereotyping by decreasing the reliance upon an individual's protected characteristic as a trivial solution, instead looking for more unique relations and providing model stability. The loss equation therefore penalises deviation in the protected attribute from its mean, encouraging the model to be fair by minimising these variations.

$$L_{\text{var}} = \max\left(0, 1 - \sqrt{\mathbb{E}\left[(P - \mathbb{E}[P])^2\right] + \varepsilon}\right) \tag{1}$$

where $P$ is the protected attribute, and $\varepsilon = 1\mathrm{e}{-4}$ to ensure numerical stability.

**Invariance**: This ensures consistent results for similar inputs, e.g. if two candidates have the same qualifications and skills, but are from different religions, this variation should not influence the decision and therefore promotes individual fairness. The loss term here should directly penalise the variance of the protected attribute, promoting invariance with respect to it.

$$L_{\text{inv}} = \mathbb{E}\left[(P - \mathbb{E}[P])^2\right] \tag{2}$$

where $P$ is the protected attribute.

**Covariance**: This aims to reduce the model's dependency on protected characteristics to make predictions, and to ensure decisions are made independently of them and promotes group fairness. The loss equation therefore minimises this covariance.

$$L_{\text{cov}} = \frac{\sqrt{\sum\left((\hat{y} - \mathbb{E}[\hat{y}])^{\top} \cdot P\right)^2}}{N} \tag{3}$$

where $\hat{y}$ is the model's prediction, $P$ is the protected attribute, and $N$ is the number of samples.

---

**Algorithm 1** FairVIC Loss Function

---

1: **Input:** Model $M$, Epochs $E$, Batch size $B$, Data $D$, Protected attribute $P$, Weights $(\lambda_{\text{acc}}, \lambda_{\text{var}}, \lambda_{\text{inv}}, \lambda_{\text{cov}})$
2: **Output:** Trained Model $M$
3: Initialise $M$
4: **for** $e \in E$ **do**
5:     Shuffle data $D$
6:     **for** each batch $\{(X, Y)\} \in D$ with size $B$ **do**
7:         $\hat{Y} \leftarrow M(X)$
8:         Calculate FairVIC Loss:
9:           $L_{\text{acc}} \leftarrow \text{AccuracyLoss}(Y, \hat{Y})$
10:        $L_{\text{var}} \leftarrow \text{VarianceLoss}(P)$
11:        $L_{\text{inv}} \leftarrow \text{InvarianceLoss}(P)$
12:        $L_{\text{cov}} \leftarrow \text{CovarianceLoss}(\hat{Y}, P)$
13:        $L_{\text{total}} \leftarrow \lambda_{\text{acc}} L_{\text{acc}} + \lambda_{\text{var}} L_{\text{var}} + \lambda_{\text{inv}} L_{\text{inv}} + \lambda_{\text{cov}} L_{\text{cov}}$
14:        Compute gradients $\nabla L_{\text{total}} \leftarrow \frac{\partial L_{\text{total}}}{\partial M}$
15:        Update model parameters $M \leftarrow M - \alpha \nabla L_{\text{total}}$

---

During the training of a deep learning model, the model iterates over epochs $E$. Data is shuffled into batches, upon which the model predicts to produce a set of predictions $\hat{Y}$. Typically, the true labels $Y$ and predictions $\hat{Y}$ are then passed into a suitable accuracy loss function (e.g., binary cross-entropy, hinge loss, Huber loss, etc.) and the resulting loss attempts to be minimised by an optimiser.

In the case of FairVIC, in addition to computing a suitable accuracy loss $L_{acc}$, we also calculate our three novel terms $L_{var}, L_{inv}$, and $L_{cov}$ using Equations 1, 2, and 3 respectively. Each of these individual loss terms is then multiplied by its respective weighting factor $\lambda$ and summed to form the total loss $L_{total}$. Subsequently, gradients are computed, and the optimiser adjusts the model parameters with respect to this combined loss. Further details are provided in Algorithm 1.

The multipliers $\lambda$ enable users to balance the trade-off between fairness and predictive performance, which is typical in bias mitigation techniques. Assigning a higher weight to $\lambda_{\text{acc}}$ directs the model to prioritise accuracy while increasing the weights of $(\lambda_{\text{var}}, \lambda_{\text{inv}}, \lambda_{\text{cov}})$ shifts the focus towards enhancing fairness in the model's predictions. In our implementation, the lambda coefficients $(\lambda_{\text{acc}}, \lambda_{\text{var}}, \lambda_{\text{inv}}, \lambda_{\text{cov}})$ are constrained such that their sum equals one. In other words, $\lambda_{\text{acc}} = 1 - \lambda_{\text{var}} - \lambda_{\text{inv}} - \lambda_{\text{cov}}$. This normalisation ensures the optimisation will not produce multiple solutions in the form $\{k.\lambda_{\text{acc}}, k.\lambda_{\text{var}}, k.\lambda_{\text{inv}}, k.\lambda_{\text{cov}}\}, k \in \mathcal{R}$.

### 4.2 MULTI-OBJECTIVE LAMBDA TUNING

To address the challenges associated with hyperparameter selection and tuning, specifically surrounding the multipliers $\lambda$ used to balance the trade-off between fairness and predictive performance, we introduce an adaptive multi-objective gradient descent extension to FairVIC aimed to dynamically adjust the $\lambda$ parameters during the models training. Optimising $\lambda$ multipliers this way, instead of on performance metrics, helps to ensure models are trainable and adaptable, addressing the differentiability issues metrics often present. Crucially, this approach also reduces the risk of human biases influencing the model, as it allows for the direct integration of fairness-focused adjustments, avoiding the unintentional reinforcement of existing disparities.

This process, seen in Algorithm 2, first involves converting the existing $\lambda$ multipliers $(\lambda_{\text{acc}}, \lambda_{\text{var}}, \lambda_{\text{inv}}, \lambda_{\text{cov}})$ to trainable variables that can be adjusted during training. To optimise these, we employ gradient descent. The gradients of the loss $L$ with respect to each $\lambda_i$ are computed as follows:

$$\frac{\partial L}{\partial \lambda_i} = \frac{\partial L_i}{\partial \lambda_i} + \frac{\partial R}{\partial \lambda_i} \tag{4}$$

where $R$ is a regularization term to penalise overfitting towards large or small values. To ensure no single $\lambda_i$ disproportionately influences the optimization process, the gradients are scaled:

---

**Algorithm 2** Multi-Objective Lambda Tuning in FairVIC

1: **Input:** Initial $\lambda$ values ($\lambda_{\text{binary}}$, $\lambda_{\text{var}}$, $\lambda_{\text{inv}}$, $\lambda_{\text{cov}}$), Learning rate $\alpha_\lambda$
2: **Output:** Optimised $\lambda$ values
3: **for** each training epoch **do**
4:     **for** each batch **do**
5:         Compute gradients $\nabla \frac{\partial L}{\partial \lambda_i} = \frac{\partial L_i}{\partial \lambda_i} + \frac{\partial R}{\partial \lambda_i}$
6:         Scale gradients $\tilde{g}_{\lambda_i} = \frac{g_{\lambda_i}}{\|g_{\lambda_i}\| + \epsilon}$
7:         Fine-tune multipliers $\lambda_i \leftarrow \lambda_i - \alpha_\lambda \tilde{g}_{\lambda_i}$

---

$$\tilde{g}_{\lambda_i} = \frac{g_{\lambda_i}}{\|g_{\lambda_i}\| + \epsilon} \tag{5}$$

where $g_i$ is the gradient of $L$ with respect to $\lambda_i$, and $\epsilon$ is a small constant to avoid division by zero. This normalization step prevents the dominance of larger gradients, promoting a more balanced and effective adjustment of the $\lambda$ values during training. Finally, the multipliers $\lambda$ are updated according to these gradients $\tilde{g}_{\lambda_i}$ and the learning rate $\alpha$ automatically.

The introduction of this multi-objective lambda tuning extension to FairVIC reduces the need for extensive manual hyperparameter selection and tuning, thereby decreasing the risk of introducing associated biases into the model. This not only streamlines model development but also effectively abstracts the complexity of manual hyperparameter optimization, thereby enhancing usability and reducing the scope for human error in the tuning process.

## 5 EXPERIMENTS

In our experimental evaluation, we assess the performance of FairVIC[1] against a set of comparable in-processing bias mitigation methods on a series of datasets known for their bias. Here, we describe the datasets used and the methods we compare against.

### 5.1 DATASETS

We evaluate FairVIC on four datasets that are used as benchmarks in bias mitigation evaluation due to their known biases towards certain subgroups of people within their sample population. These datasets allow for highlighting the generalisable capabilities of FairVIC across different demographic disparities.

**Tabular datasets.** The main body of evaluation is done using three tabular datasets: Adult Income (Becker & Kohavi, 1996), COMPAS (Angwin et al., 2022), and German Credit (Hofmann, 1994), all of which are binary classification tasks. Adult Income aims to predict whether an individual's income is $> \$50K$ or $\leq \$50K$. It is known for its gender and racial biases in economic disparity. The Correctional Offender Management Profiling for Alternative Sanctions (COMPAS) dataset is frequently used for evaluating debiasing techniques. It has a classification goal of predicting recidivism risks and is infamous for its racial biases. Finally, the German Credit dataset was used to assess creditworthiness by classifying individuals as either good or bad credit risks, with known biases related to age and gender (Kamiran & Calders, 2009).
**Language dataset.** To show the ability of FairVIC to work for different data modalities, we also utilise CivilComments-WILDS – a natural language dataset. We take a sample of 50K comments from the dataset, which is comprised of a collection of comments on the Civil Comments platform. The binary classification goal is to label comments as toxic or non-toxic. Our sample is stratified to retain the same proportion of toxic comments as in the original dataset. We take ethnicity as the protected characteristic where comments are marked as white or non-white (Koh et al., 2021).

Detailed metadata for each dataset, including our selections for protected groups, can be found in Appendix A.1.

---

[1]The code for our FairVIC implementation is available at: `https://anonymous.4open.science/r/FairVIC-BEE7`

## 5.2 Comparable Techniques

To highlight the performance of FairVIC, we evaluate against five comparable in-processing bias mitigation methods. These are:

**Adversarial Debiasing.** This method leverages an adversarial network that aims to predict protected characteristics based on the predictions of the main model. The primary model seeks to maximise its own prediction accuracy while minimising the adversary's prediction accuracy (Zhang et al., 2018).

**Exponentiated Gradient Reduction.** This technique reduces fair classification to a sequence of cost-sensitive classification problems, returning a randomised classifier with the lowest empirical error subject to a chosen fairness constraint (Agarwal et al., 2018).

**Meta Fair Classifier.** This classifier allows a fairness metric as an input and optimises the model with respect to regular performance and the chosen fairness metric (Celis et al., 2019).

**Fair MixUp.** This technique generates synthetic samples by linearly interpolating between pairs of training data points by protected attribute to smooth decision boundaries. The loss function is then further constrained by a fairness metric (Chuang & Mroueh, 2021).

**FairGBM.** This method uses a gradient-boosting decision tree model that integrates fairness constraints directly into the boosting process by adjusting the loss function to account for fairness metrics (Cruz et al., 2023).

Alongside this, a baseline-biased model was established that was a neural network with binary cross-entropy loss only. Details on the neural network architecture/ hyperparameters used for both the baseline model and the FairVIC model can be found in Appendix A.2.

## 6 Evaluation

### 6.1 Core Results Analysis

To assess the prediction and fairness performance of FairVIC[2] and state-of-the-art approaches, we test all methods across each tabular dataset to enable a fair comparison. Table 1 shows these results. We have also provided Figure 1, which visualises the absolute difference from the ideal value of each metric, highlighting how far each method deviates from perfect accuracy and fairness on the Adult Income dataset. For the COMPAS and German Credit datasets, similar figures can be found in Appendix B.1 in Figures 3 and 4.

Across all three datasets, the baseline performs poorly in fairness but obtains higher performance scores, which is expected. For example, in the Adult Income dataset, the baseline model shows a relatively high accuracy (0.8517), while exhibiting poor fairness with regard to Disparate Impact (0.3278). The baseline highlights the need for a bias mitigation approach that works across all metrics simultaneously, as the low bias in terms of Equalised Odds (0.0810) and Absolute Odds (0.0662) alone could misleadingly suggest that the model is fair, rather than that the bias needs to be captured differently. This has the effect that approaches requiring a single fairness constraint, such as Exponentiated Gradient Reduction, often leave significant bias unaddressed in the data.

Overall, FairVIC outperforms all other comparable methods by demonstrating consistent improvements in both fairness and accuracy retention. As seen in Figure 1, our FairVIC model achieves the lowest cumulative absolute error from perfect accuracy and fairness in the Adult Income dataset, effectively balancing the fairness-accuracy trade-off. The trend is consistent across the COMPAS and German Credit datasets as well. FairVIC is only second to Fair MixUp in the German Credit dataset by an absolute difference of $\approx 0.0271$, perhaps due to Fair MixUp's approach coincidentally capturing the specific type of bias that is present in this data only. Comparatively, in the Adult Income and COMPAS datasets Fair MixUp performs poorly, ranking seventh and third place respectively in terms of absolute differences. This again exemplifies the ability of FairVIC's approach to generalise across datasets, making it a more versatile solution.

Other comparable methods are generally not as effective as FairVIC, each exhibiting different shortcomings. For instance, MetaFair never improves upon the baseline in cumulative absolute difference from the ideal value, while Exponentiated Gradient Reduction struggles to balance the improve-

---

[2]Multi-objective lambda recommendations were applied to FairVIC to obtain the results in Table 1, see Section 6.4

Table 1: FairVIC accuracy and fairness results, compared with the base biased model, and five other comparable methods for bias mitigation in-processing for each of the three tabular datasets.

| Dataset | Model | Accuracy | F1 Score | Equalized Odds | Absolute Odds | Demographic Parity | Disparate Impact |
|---|---|---|---|---|---|---|---|
| Adult Income | Baseline (Biased) | 0.8517 ± 0.0021 | 0.6475 ± 0.0214 | 0.0810 ± 0.0308 | 0.0662 ± 0.0219 | -0.1580 ± 0.0218 | 0.3278 ± 0.0271 |
| | Adversarial Debiasing | 0.8065 ± 0.0048 | 0.4773 ± 0.0708 | 0.2127 ± 0.0828 | 0.1172 ± 0.0443 | -0.0405 ± 0.0679 | 0.7874 ± 0.2185 |
| | Exponentiated Gradient Reduction | 0.8027 ± 0.0026 | 0.4056 ± 0.0052 | 0.0238 ± 0.0115 | 0.0167 ± 0.0061 | -0.0601 ± 0.0026 | 0.4602 ± 0.0237 |
| | Meta Fair Classifier | 0.5171 ± 0.0602 | 0.4744 ± 0.0219 | 0.4826 ± 0.0894 | 0.2935 ± 0.0497 | -0.2098 ± 0.0542 | 0.7140 ± 0.0812 |
| | Fair MixUp | 0.7755 ± 0.0098 | 0.4565 ± 0.0414 | 0.1395 ± 0.0795 | 0.1074 ± 0.0568 | -0.1368 ± 0.0524 | 0.3921 ± 0.1654 |
| | FairGBM | 0.8731 ± 0.0026 | 0.7122 ± 0.0079 | 0.0658 ± 0.0131 | 0.0583 ± 0.0092 | -0.1707 ± 0.0044 | 0.3363 ± 0.0151 |
| | FairVIC | 0.8306 ± 0.0078 | 0.5484 ± 0.0492 | 0.2812 ± 0.0392 | 0.1539 ± 0.0251 | -0.0136 ± 0.0243 | 0.9407 ± 0.1569 |
| COMPAS | Baseline (Biased) | 0.6619 ± 0.0175 | 0.6280 ± 0.0206 | 0.3095 ± 0.0744 | 0.2540 ± 0.0683 | -0.2889 ± 0.0651 | 0.6085 ± 0.0853 |
| | Adversarial Debiasing | 0.6581 ± 0.0185 | 0.6253 ± 0.0124 | 0.1707 ± 0.0694 | 0.1363 ± 0.0504 | -0.0902 ± 0.1367 | 0.8982 ± 0.2614 |
| | Exponentiated Gradient Reduction | 0.5574 ± 0.0169 | 0.2981 ± 0.0407 | 0.0630 ± 0.0333 | 0.0432 ± 0.0231 | -0.0393 ± 0.0257 | 0.9545 ± 0.0293 |
| | Meta Fair Classifier | 0.3471 ± 0.0147 | 0.4312 ± 0.0380 | 0.2951 ± 0.1038 | 0.2257 ± 0.1095 | 0.2526 ± 0.1070 | 2.5876 ± 0.6627 |
| | Fair MixUp | 0.6012 ± 0.0234 | 0.5502 ± 0.0443 | 0.1225 ± 0.0763 | 0.0883 ± 0.0493 | -0.0680 ± 0.0808 | 0.8964 ± 0.1242 |
| | FairGBM | 0.6440 ± 0.0151 | 0.6254 ± 0.0153 | 0.2015 ± 0.1128 | 0.1466 ± 0.0961 | 0.0881 ± 0.1225 | 1.2828 ± 0.4058 |
| | FairVIC | 0.6522 ± 0.0216 | 0.6079 ± 0.0374 | 0.0867 ± 0.0401 | 0.0571 ± 0.0270 | -0.0294 ± 0.0554 | 0.9598 ± 0.0850 |
| German Credit | Baseline (Biased) | 0.7325 ± 0.0232 | 0.8170 ± 0.0199 | 0.2101 ± 0.0871 | 0.1464 ± 0.0685 | -0.1728 ± 0.1009 | 0.7860 ± 0.1241 |
| | Adversarial Debiasing | 0.5815 ± 0.1513 | 0.6302 ± 0.2581 | 0.1020 ± 0.0418 | 0.0737 ± 0.0404 | -0.0657 ± 0.0335 | 0.8084 ± 0.2130 |
| | Exponentiated Gradient Reduction | 0.7465 ± 0.0300 | 0.8321 ± 0.0208 | 0.1232 ± 0.0631 | 0.0796 ± 0.0348 | -0.1084 ± 0.0746 | 0.8692 ± 0.0896 |
| | Meta Fair Classifier | 0.7575 ± 0.0260 | 0.8291 ± 0.0229 | 0.2215 ± 0.1112 | 0.1444 ± 0.0810 | -0.1052 ± 0.1315 | 0.8601 ± 0.1755 |
| | Fair MixUp | 0.6865 ± 0.0426 | 0.7740 ± 0.0381 | 0.1359 ± 0.0692 | 0.0925 ± 0.0422 | -0.0189 ± 0.0728 | 0.9781 ± 0.1082 |
| | FairGBM | 0.7460 ± 0.0348 | 0.8255 ± 0.0283 | 0.1922 ± 0.0906 | 0.1345 ± 0.0756 | -0.1539 ± 0.0773 | 0.8081 ± 0.0915 |
| | FairVIC | 0.7200 ± 0.0383 | 0.8110 ± 0.0342 | 0.1914 ± 0.0718 | 0.1367 ± 0.0581 | 0.0140 ± 0.1289 | 1.0247 ± 0.1653 |

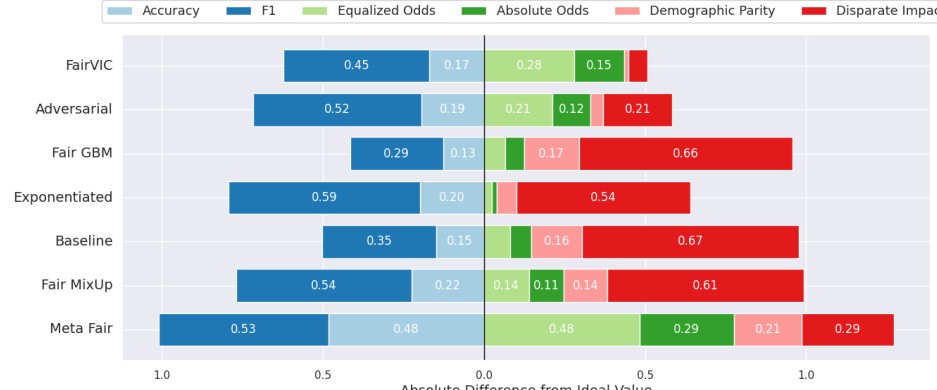

Figure 1: Absolute differences from the ideal value (e.g., perfect accuracy and fairness) in performance (left) and fairness (right) metrics of comparable techniques, sorted in ascending order, on the Adult Income dataset.

ments across all fairness metrics, often prioritising Equalised and Absolute Odds over Disparate Impact, particularly in the Adult Income dataset. Similarly, the Adversarial Debiasing model, though initially promising and achieving second place after FairVIC in the Adult Income and COMPAS datasets, fails to maintain its performance on the German Credit dataset, where its results fall below even the baseline.

Overall, FairVIC's ability to consistently balance the trade-off between fairness and accuracy, adapt to various datasets, and handle all fairness metrics comprehensively makes it the most effective method. Its consistent performance across different datasets, as evidenced by the lowest cumulative absolute error in performance and fairness, solidifies its superiority over other comparable methods.

## 6.2 INDIVIDUAL FAIRNESS ANALYSIS

To emphasise further FairVIC's ability to perform well across all fairness metrics, we also evaluate upon individual fairness by outputting the results of the counterfactual model, as described in Section 3.3. The full results, alongside the absolute difference in averages for each metric across the regular and counterfactual models, are seen in Table 6 in Appendix B.3.

The FairVIC model shows considerable promise in enhancing individual fairness across different datasets when compared with the baseline models. The counterfactual results from the FairVIC model generally exhibit lower absolute differences in metrics, particularly in the COMPAS and

German Credit datasets. For example, in the German Credit dataset, the mean absolute difference across all six metrics between the regular and the counterfactual baseline model is $\approx 0.0348$, while for FairVIC's regular and counterfactual models it is lower at $\approx 0.0106$. This suggests a more stable and fair performance under counterfactual conditions. This capability highlights FairVIC's strength in not only addressing group fairness but also ensuring that individual decisions remain consistent and fair when hypothetical scenarios are considered.

## 6.3 LAMBDA ABLATION STUDY ANALYSIS

The FairVIC loss terms are combined with binary cross entropy for training the NN to enable optimisation of both accuracy and fairness, minimising the trade-off. The effect of FairVIC on the overall loss function can be increased and decreased by changing the weight $\lambda$ for each FairVIC term. To evaluate this effect, we train a number of neural networks with the architecture described in Appendix A.2, with a different $\lambda_{\text{acc}}$ weighting each time. In this initial experiment, we evaluate the effect of weighting the FairVIC loss terms equally, so that $\lambda_{\text{var}} = \lambda_{\text{inv}} = \lambda_{\text{cov}} = \frac{(1-\lambda_{\text{acc}})}{3}$, where $0 < \lambda_{\text{acc}} < 1$. The performance and fairness measures for each model are listed in Table 7 in Appendix C, and visualisations for the absolute difference in performance and fairness from ideal values for each run are visualised in Figure 2.

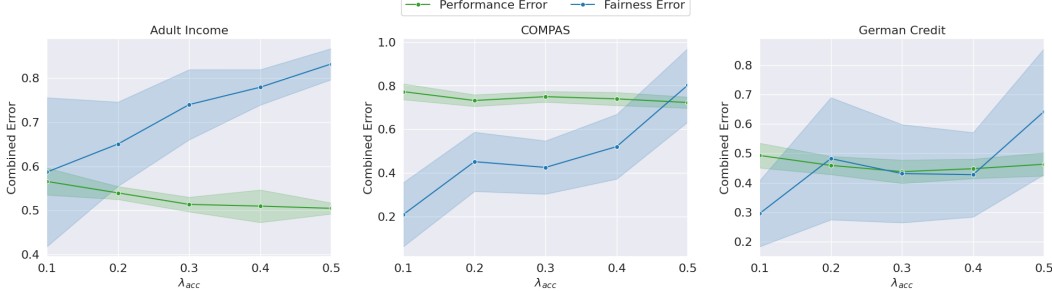

Figure 2: Absolute difference from the ideal value for performance (green) and fairness (blue) metrics of FairVIC with varying $\lambda_{\text{acc}}$ values across all tabular datasets. The FairVIC terms are weighted equally, such that $\lambda_{\text{acc}} + \lambda_{\text{var}} + \lambda_{\text{inv}} + \lambda_{\text{cov}} = 1$.

In Figure 2, the trade-off between accuracy and fairness is evident. As $\lambda_{\text{acc}}$ increases, predictive performance improves, but the fairness metrics deviate further from the ideal value. In contrast, when $\lambda_{\text{acc}}$ is lower, fairness improves, but this time with only a negligible drop in accuracy. This suggests that lower $\lambda_{\text{acc}}$ values provide a better overall performance balance.

To evaluate upon the effect of each individual VIC term within the loss function, we can suppress the lambda terms from two out of three of variance, invariance, and covariance to leave only one remaining. We keep $\lambda_{\text{acc}} = 0.1$ since the previous lambda experiment showed this to be most effective, while the chosen FairVIC loss term is given a weighting of 0.9. The performance and fairness results for each experiment with different weightings are listed in Table 8.

It can be concluded that each term has a different effect. Variance is shown to have the lowest standard deviation across all metrics and all data in Table 8, offering stability to FairVIC. Utilising only the covariance term has the greatest effect on group fairness, as seen in Table 8. The effect of only adding weighting to the invariance term can be assessed using Table 6. The invariance term aims to give similar outputs to similar inputs, therefore it should have more of an effect towards individual fairness. Table 6 corroborates this hypothesis, as the FairVIC Individual model (FairVIC with only individual loss weighted to 0.9) consistently has a lower absolute difference than the baseline between the regular and counterfactual models across all metrics and tabular datasets, signalling greater individual fairness. Therefore, we conclude that the combination of all three terms would aim to improve both group and individual fairness, and increase stability.

Table 2: Lambda recommendations provided by the Multi-Objective Lambda tuning extension, for all four datasets.

| Dataset | Recommended $\lambda$ | | | |
| --- | --- | --- | --- | --- |
| | $\lambda_{\text{acc}}$ | $\lambda_{\text{var}}$ | $\lambda_{\text{inv}}$ | $\lambda_{\text{cov}}$ |
| Adult Income | 0.10 | 0.10 | 0.10 | 0.70 |
| COMPAS | 0.10 | 0.10 | 0.10 | 0.70 |
| German Credit | 0.10 | 0.10 | 0.10 | 0.70 |
| CivilComments-WILDS | 0.10 | 0.10 | 0.15 | 0.65 |

## 6.4 MULTI-OBJECTIVE LAMBDA TUNING ANALYSIS

To extend FairVIC, we propose a Multi-Objective Lambda Tuning Algorithm as described in Section 4.2. Upon application to all four datasets, the following results were outputted as seen in Table 2. Each individual $\lambda$ was weighted equally ($= 0.25$) at the start of training and converged to the recommended values.

For all datasets, a recommendation of $\lambda_{\text{acc}}$ of 0.10 is given. This is the same conclusion that was drawn through our trade-off study, in Section 6.3, as the lower the $\lambda_{\text{acc}}$, the higher the fairness. Out of the three VIC loss terms, covariance is assigned with the highest weighting. This corroborates the findings of the ablation study in Table 8, where covariance held the heaviest weighting on group fairness results. Variance and invariance are recommended weights of $0.10$ to $0.15$ for use for all datasets. Both of these components being lower is to be expected, due to their low effect on group fairness. However, as discussed in Section 6.3, both of these terms hold a positive effect through variance offering stability and invariance improving individual fairness. Therefore, the combination of all four terms with these weights allows for effective results throughout in the FairVIC model, as seen in the final results in Table 1.

## 6.5 LANGUAGE DATASET RESULTS

To show FairVIC's versatility across data modalities, our approach was applied to the CivilComments-WILDS dataset. The results are shown in Table 3, where FairVIC uses the lambda recommendations as shown in Section 6.4.

From Table 3, the same trend can be seen as in the tabular dataset results, where FairVIC gives fairer results across all tested fairness metrics, the most notable being the improvement to disparate impact from 1.2069 to 1.0087. The model also shows good stability throughout. Therefore, FairVIC is not confined to one modality. The use of a different model architecture also proves FairVIC's adaptability to be utilised within different neural networks.

Table 3: FairVIC and baseline comparison results of both performance and fairness for the CivilComments-WILDS dataset.

| Model | Accuracy | F1 Score | Equalized Odds | Absolute Odds | Demographic Parity | Disparate Impact |
| --- | --- | --- | --- | --- | --- | --- |
| Baseline | $0.8993 \pm 0.0026$ | $0.4094 \pm 0.0448$ | $0.1491 \pm 0.0363$ | $0.1198 \pm 0.0257$ | $0.1621 \pm 0.0257$ | $1.2069 \pm 0.0430$ |
| FairVIC | $0.8962 \pm 0.0038$ | $0.2813 \pm 0.1142$ | $0.0854 \pm 0.0495$ | $0.0455 \pm 0.0260$ | $0.0082 \pm 0.0082$ | $1.0087 \pm 0.0088$ |

## 7 CONCLUSION AND FUTURE WORK

In this paper, we introduced FairVIC, an in-processing bias mitigation technique that introduces three new terms into the loss function of a neural network- variance, invariance, and covariance. Across our experimental evaluation, FairVIC significantly improves scores for all fairness metrics compared to previous comparable methods which typically aim to improve only upon a single metric. This balance showcases FairVIC's strength in providing a robust and generalisable solution applicable across various tasks and datasets. We also presented an extension to automatically tune the weights of each term during training and conducted a full ablation study to analyse the effect of each term in FairVIC. Future work would look to extend FairVIC to consider multiple protected characteristics simultaneously and expand its utility to image datasets.

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

## A EXPERIMENT DETAILS

### A.1 DATASET METADATA

Detailed metadata for each dataset, including our selection of privileged group, can be found in Table 4. Note that for CivilComments-WILDS, the number of features is obtained by combining the protected characteristic and the toxicity label with the 50 tokenised text features.

Table 4: Metadata on all four experimental datasets.

| Dataset | Adult Income | COMPAS | German Credit | CivilComments-WILDS |
|---|---|---|---|---|
| No. of Features | 11 | 8 | 20 | 52 |
| No. of Rows | 48,842 | 5,278 | 1,000 | 50,000 |
| Target Variable | income | two_year_recid | credit | toxicity |
| Favourable Label | >50K (1) | False (0) | Good (1) | Non-Toxic (0) |
| Unfavourable Label | <=50K (0) | True (1) | Bad (0) | Toxic (1) |
| Protected Characteristic | sex | race | age | race |
| Privileged Group | male (1) | Caucasian (1) | >25 (1) | white (1) |
| Unprivileged Group | female (0) | African-American (0) | <=25 (0) | non-white (0) |

### A.2 NEURAL NETWORK CONFIGURATION

The configurations for the neural networks utilised for both the tabular and language data can be seen in Table 5. It should be noted that the language dataset architecture also included scheduled learning rate reduction. To obtain results, each model was run 10 times over random seeds, with a randomised train/test split each time. The averages and standard deviations were the outputted from across all 10 runs.

Table 5: Experimental model setup and parameters.

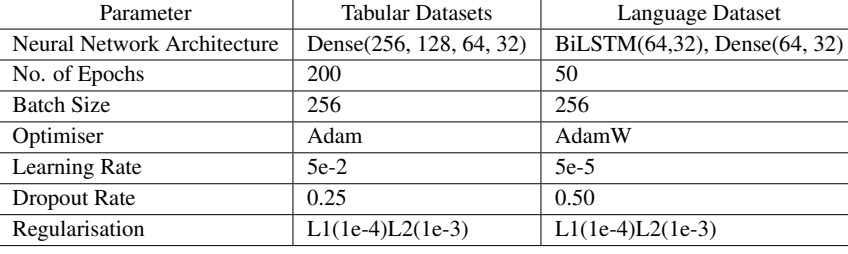

| Parameter | Tabular Datasets | Language Dataset |
|---|---|---|
| Neural Network Architecture | Dense(256, 128, 64, 32) | BiLSTM(64,32), Dense(64, 32) |
| No. of Epochs | 200 | 50 |
| Batch Size | 256 | 256 |
| Optimiser | Adam | AdamW |
| Learning Rate | 5e-2 | 5e-5 |
| Dropout Rate | 0.25 | 0.50 |
| Regularisation | L1(1e-4)L2(1e-3) | L1(1e-4)L2(1e-3) |

All models were run with minimal and consistent data preprocessing. While some models, such as MetaFair, may underperform due to their reliance on specific sampling techniques, all comparable methods are treated uniformly as in-processing techniques. This allows them to be applied out-of-the-box to any dataset, ensuring a fair evaluation across models.

## B FULL TRAINING RESULTS

In addition to the results and analysis presented in Section 6, this section provides supplementary experiments and figures.

### B.1 RESULTS VISUALISATIONS FOR THE COMPAS AND GERMAN CREDIT DATASET

Following the analysis discussed in Section 6.1 regarding Figure 1, the absolute difference from the ideal value for the COMPAS and German Credit datasets can be visualised in Figures 3 and 4 respectively. Analysis regarding these additional figures can be found in Section 6.1.

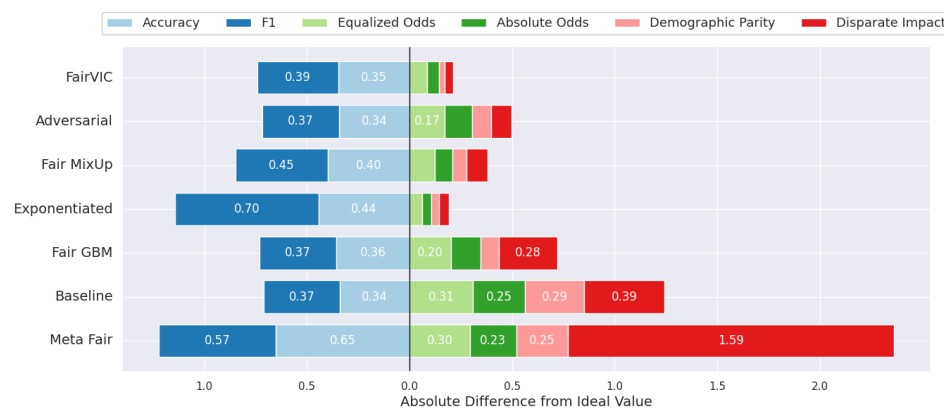

Figure 3: Absolute difference from the ideal value in performance (left) and fairness (right) metrics of comparable techniques, sorted in ascending order, on the COMPAS dataset.

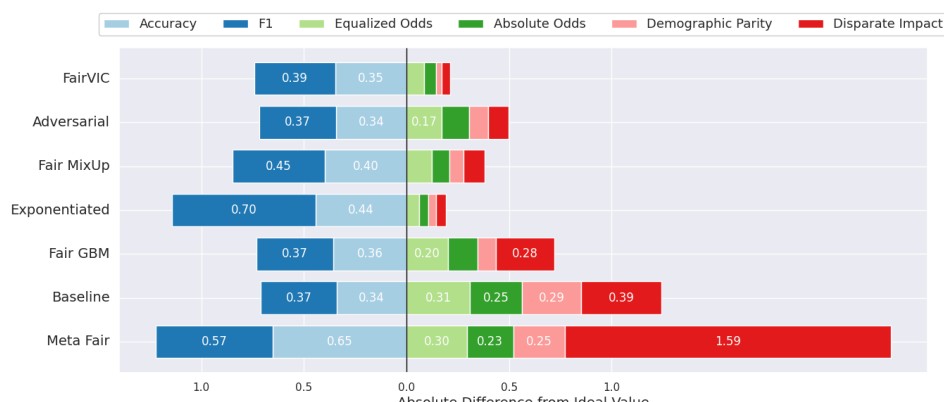

Figure 4: Absolute difference from the ideal value in performance (left) and fairness (right) metrics of comparable techniques, sorted in ascending order, on the German Credit dataset.

## B.2 FEATURE IMPORTANCES

Figure 5 shows the feature importance of the baseline and FairVIC models across three tabular datasets. In all baseline models, the protected attributes show some importance to the decision-making process, such as in the COMPAS dataset, where *race* is a dominant feature. Combined with the results presented in Section 6.1, this suggests that the baseline models are prone to using the protected attribute to propagate bias. Additionally, proxy variables (highlighted with their importance in black), which are strongly correlated with the protected attributes, further show how bias can be perpetuated in the baseline model. For example, in the Adult Income dataset, *relationship* has a mean feature importance of $0.0124$. This indicates that even though the model appears to have limited reliance on the protected attribute *sex* (which is among the least used features), it may still propagate bias through proxies such as *relationship*.

In contrast, the FairVIC models for all three datasets demonstrate a strong reduction in the mean importance of protected attributes and proxy variables. This reduction is due to the three additional terms used in FairVIC- variance, invariance, and covariance. We can see that the covariance term exactly minimises the model's dependency on the protected characteristic, which, in combination with results in Section 6.1, suggests a fairer decision-making process. The reduction in proxy variables should also be noted. Not only does FairVIC successfully reduce the reliance on the protected attribute, but it can also reduce the reliance on any features strongly correlated to the protected attribute. For example, in the Adult Income dataset, *sex* and *relationship* have a strong negative correlation ($-0.58$) meaning a model cannot only propagate bias through the use of *sex* but also

through the use of *relationship* which we see the baseline model rely upon. The FairVIC model sees the mean feature importance of *relationship* drop by approximately a third and the importance of *sex* drop by half. This shows FairVIC's ability to mitigate both direct and indirect biases, leading to more equitable outcomes.

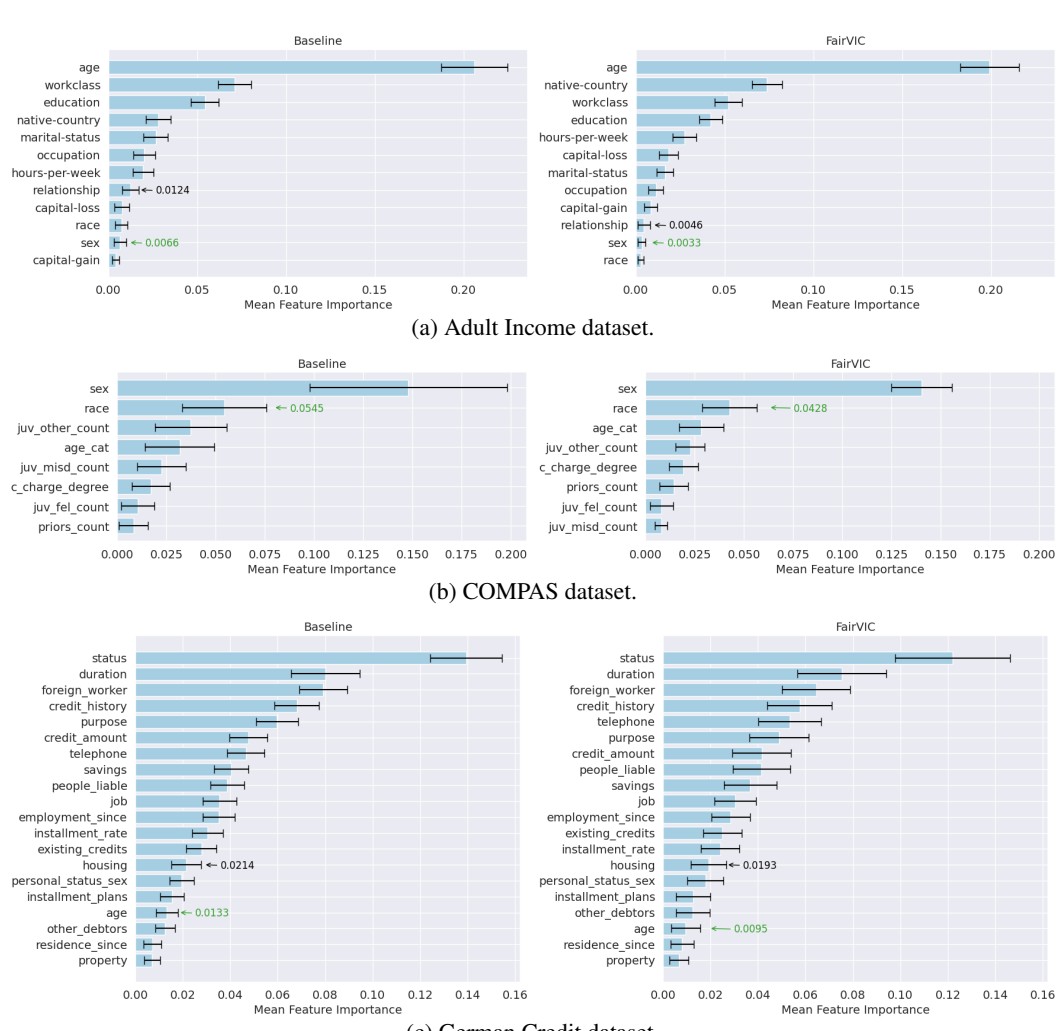

(a) Adult Income dataset.

(b) COMPAS dataset.

(c) German Credit dataset.

Figure 5: Mean feature importances for the baseline and FairVIC models across three tabular datasets. The protected attribute (green) and strong proxy variables to the protected attribute (black) are annotated for their exact feature importance.

## B.3 INDIVIDUAL FAIRNESS RESULTS

Following the analysis found in Section 6.2, Table 6 shows the individual fairness on both the baseline and FairVIC models using their counterfactual model results. In the Adult Income dataset, the mean absolute difference across all six metrics combined for the baseline model is $\approx 0.0034$, while for FairVIC it is $\approx 0.0064$. In the COMPAS dataset, the mean absolute difference across all six metrics combined for the baseline model is $\approx 0.013$, while for FairVIC it is $\approx 0.0050$. FairVIC improves individual fairness against the baseline model in two of the three tabular datasets and consistently achieves a significantly low mean, further highlighting the effectiveness of FairVIC.

For discussion on the FairVIC Individual model individual fairness results, see Section 6.2.

Table 6: Counterfactual (CF) model results and absolute differences (ADs) for the baseline, FairVIC ($\lambda_{\text{acc, var, inv}} = 0.1, \lambda_{\text{cov}} = 0.7$), and FairVIC Invariance ($\lambda_{\text{acc}} = 0.1, \lambda_{\text{inv}} = 0.9, \lambda_{\text{var, cov}} = 0.0$) models.

| Dataset | Model | Accuracy | F1 Score | Equalized Odds | Absolute Odds | Demographic Parity | Disparate Impact |
|---|---|---|---|---|---|---|---|
| Adult Income | Baseline | $0.8517 \pm 0.0021$ | $0.6475 \pm 0.0214$ | $0.0810 \pm 0.0308$ | $0.0662 \pm 0.0219$ | $-0.1580 \pm 0.0218$ | $0.3278 \pm 0.0271$ |
| | Baseline CF | $0.8507 \pm 0.0032$ | $0.6451 \pm 0.0237$ | $0.0767 \pm 0.0266$ | $0.0627 \pm 0.0226$ | $-0.1561 \pm 0.0220$ | $0.3353 \pm 0.0433$ |
| | Baseline AD | $0.0010 \pm 0.0038$ | $0.0024 \pm 0.0319$ | $0.0043 \pm 0.0407$ | $0.0035 \pm 0.0315$ | $0.0019 \pm 0.0310$ | $0.0075 \pm 0.0511$ |
| | FairVIC Invariance | $0.8498 \pm 0.0039$ | $0.6506 \pm 0.0229$ | $0.0959 \pm 0.0314$ | $0.0820 \pm 0.0272$ | $-0.1715 \pm 0.0295$ | $0.3164 \pm 0.0274$ |
| | FairVIC Invariance CF | $0.8487 \pm 0.0055$ | $0.6522 \pm 0.0204$ | $0.0967 \pm 0.0360$ | $0.0837 \pm 0.0317$ | $-0.1742 \pm 0.0333$ | $0.3224 \pm 0.0245$ |
| | FairVIC Invariance AD | $0.0011 \pm 0.0067$ | $0.0016 \pm 0.0307$ | $0.0008 \pm 0.0478$ | $0.0017 \pm 0.0418$ | $0.0027 \pm 0.0445$ | $0.0060 \pm 0.0368$ |
| | FairVIC | $0.8306 \pm 0.0078$ | $0.5484 \pm 0.0492$ | $0.2812 \pm 0.0392$ | $0.1539 \pm 0.0251$ | $-0.0136 \pm 0.0243$ | $0.9407 \pm 0.1569$ |
| | FairVIC CF | $0.8323 \pm 0.0052$ | $0.5641 \pm 0.0390$ | $0.2781 \pm 0.0481$ | $0.1541 \pm 0.0303$ | $-0.0155 \pm 0.0257$ | $0.9246 \pm 0.1437$ |
| | FairVIC AD | $0.0017 \pm 0.0094$ | $0.0157 \pm 0.0628$ | $0.0031 \pm 0.0621$ | $0.0002 \pm 0.0393$ | $0.0019 \pm 0.0354$ | $0.0161 \pm 0.2128$ |
| COMPAS | Baseline | $0.6619 \pm 0.0175$ | $0.6280 \pm 0.0206$ | $0.3095 \pm 0.0744$ | $0.2540 \pm 0.0683$ | $-0.2889 \pm 0.0651$ | $0.6085 \pm 0.0853$ |
| | Baseline CF | $0.6560 \pm 0.0094$ | $0.5989 \pm 0.0287$ | $0.3240 \pm 0.0655$ | $0.2584 \pm 0.0568$ | $-0.2892 \pm 0.0571$ | $0.6323 \pm 0.0714$ |
| | Baseline AD | $0.0059 \pm 0.0199$ | $0.0291 \pm 0.0353$ | $0.0145 \pm 0.0991$ | $0.0044 \pm 0.0866$ | $0.0003 \pm 0.0866$ | $0.0238 \pm 0.1112$ |
| | FairVIC Invariance | $0.6598 \pm 0.0118$ | $0.6252 \pm 0.0212$ | $0.2858 \pm 0.0810$ | $0.2402 \pm 0.0794$ | $-0.2776 \pm 0.0805$ | $0.6202 \pm 0.0969$ |
| | FairVIC Invariance CF | $0.6584 \pm 0.0155$ | $0.6133 \pm 0.0272$ | $0.2724 \pm 0.0892$ | $0.2222 \pm 0.0762$ | $-0.2583 \pm 0.0756$ | $0.6528 \pm 0.0887$ |
| | FairVIC Invariance AD | $0.0014 \pm 0.0195$ | $0.0119 \pm 0.0345$ | $0.0134 \pm 0.1205$ | $0.0180 \pm 0.1100$ | $0.0193 \pm 0.1104$ | $0.0326 \pm 0.1314$ |
| | FairVIC | $0.6522 \pm 0.0216$ | $0.6079 \pm 0.0374$ | $0.0867 \pm 0.0401$ | $0.0571 \pm 0.0270$ | $-0.0294 \pm 0.0554$ | $0.9598 \pm 0.0850$ |
| | FairVIC CF | $0.6444 \pm 0.0171$ | $0.6096 \pm 0.0481$ | $0.0983 \pm 0.0400$ | $0.0608 \pm 0.0250$ | $-0.0269 \pm 0.0587$ | $0.9571 \pm 0.1087$ |
| | FairVIC AD | $0.0078 \pm 0.0275$ | $0.0017 \pm 0.0609$ | $0.0116 \pm 0.0566$ | $0.0037 \pm 0.0368$ | $0.0025 \pm 0.0807$ | $0.0027 \pm 0.1380$ |
| German Credit | Baseline | $0.7325 \pm 0.0232$ | $0.8170 \pm 0.0199$ | $0.2101 \pm 0.0871$ | $0.1464 \pm 0.0685$ | $-0.1728 \pm 0.1009$ | $0.7860 \pm 0.1241$ |
| | Baseline CF | $0.7315 \pm 0.0505$ | $0.8158 \pm 0.0406$ | $0.1432 \pm 0.0926$ | $0.1095 \pm 0.0669$ | $-0.1246 \pm 0.0831$ | $0.8403 \pm 0.1092$ |
| | Baseline AD | $0.0010 \pm 0.0556$ | $0.0012 \pm 0.0452$ | $0.0669 \pm 0.1271$ | $0.0369 \pm 0.0957$ | $0.0482 \pm 0.1307$ | $0.0543 \pm 0.1653$ |
| | FairVIC Invariance | $0.7125 \pm 0.0238$ | $0.8003 \pm 0.0203$ | $0.1665 \pm 0.0977$ | $0.1268 \pm 0.0682$ | $-0.1191 \pm 0.0864$ | $0.8450 \pm 0.1129$ |
| | FairVIC Invariance CF | $0.7070 \pm 0.0263$ | $0.7929 \pm 0.0211$ | $0.2022 \pm 0.0645$ | $0.1411 \pm 0.0381$ | $-0.1450 \pm 0.0620$ | $0.8103 \pm 0.0761$ |
| | FairVIC Invariance AD | $0.0055 \pm 0.0355$ | $0.0074 \pm 0.0293$ | $0.0357 \pm 0.1171$ | $0.0143 \pm 0.0781$ | $0.0259 \pm 0.1063$ | $0.0347 \pm 0.1362$ |
| | FairVIC | $0.7200 \pm 0.0383$ | $0.8110 \pm 0.0342$ | $0.1914 \pm 0.0718$ | $0.1367 \pm 0.0581$ | $0.0140 \pm 0.1289$ | $1.0247 \pm 0.1653$ |
| | FairVIC CF | $0.7280 \pm 0.0214$ | $0.8184 \pm 0.0158$ | $0.1965 \pm 0.0818$ | $0.1265 \pm 0.0520$ | $0.0316 \pm 0.0672$ | $1.0404 \pm 0.0847$ |
| | FairVIC AD | $0.0080 \pm 0.0439$ | $0.0074 \pm 0.0377$ | $0.0051 \pm 0.1088$ | $0.0102 \pm 0.0780$ | $0.0176 \pm 0.1454$ | $0.0157 \pm 0.1857$ |

## B.4 MODEL REPRESENTATION ANALYSIS

An example latent space visualisation from the baseline model and FairVIC can be seen in Figure 6. In the baseline model, we observe a separation between subgroups, where women (subgroup 0) are predominantly located in the upper region and men (subgroup 1) in the lower region of the latent space. This separation suggests that the baseline model's representations may be influenced by the protected attribute, leading to the biased decision-making reported in Table 1. In contrast, the FairVIC model shows a more condensed and overlapping distribution of both subgroups within the same latent space. This indicates, alongside results in Table 1 and feature importance in Figure 5a that FairVIC has successfully reduced the model's reliance on the protected characteristic and any proxy variables, thereby promoting more equitable representations. The overlapping and compact structure in the FairVIC latent space demonstrates that similar data points, regardless of their subgroup membership, are mapped closer together, ensuring that the model's predictions are not unfairly biased towards one group over the other.

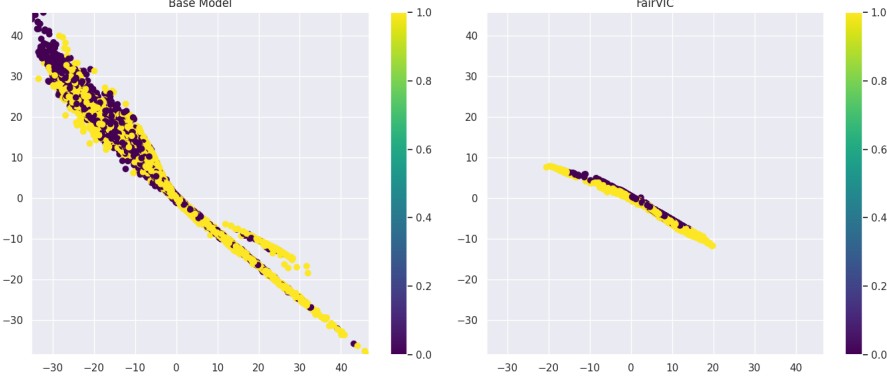

Figure 6: An example latent space visualization from one random seed of a baseline model and a FairVIC model on the Adult Income dataset. Subgroup (1) represents male individuals, and subgroup (0) represents female individuals.

### B.5 Model Optimization Analysis

Figure 7 illustrates the loss landscapes of the baseline and FairVIC models on the Adult Income dataset. Both models exhibit smooth loss surfaces, indicating that they are relatively well-optimized. The baseline model (left) shows a stable loss landscape with a slight gradient. The FairVIC model (right), despite incorporating additional fairness constraints, maintains a similarly smooth surface. This demonstrates that the inclusion of variance, invariance, and covariance terms in the loss function does not introduce instability or optimization challenges.

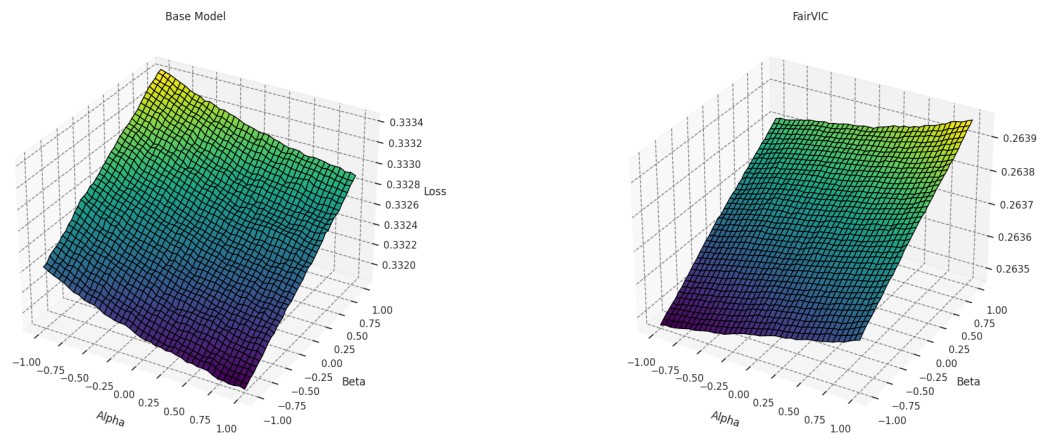

Figure 7: An example loss landscape visualisation from one random seed of a baseline model and a FairVIC model on the Adult Income dataset.

### B.6 Convergence Analysis

Alongside showing that the FairVIC model is relatively well-optimized, in this section we establish the conditions under which the FairVIC loss function converges. The full FairVIC loss function is defined as:

$$L_{\text{total}} = \lambda_{\text{acc}} L_{\text{acc}} + \lambda_{\text{var}} L_{\text{var}} + \lambda_{\text{inv}} L_{\text{inv}} + \lambda_{\text{cov}} L_{\text{cov}} \tag{6}$$

where each term is non-negative and is weighted by a corresponding $\lambda$ such that $\lambda_{\text{acc}} + \lambda_{\text{var}} + \lambda_{\text{inv}} + \lambda_{\text{cov}} = 1$. To consider convergence, we first consider the differentiability of each component in Equation 6. $L_{\text{acc}}$, in our experiments, is represented as binary cross-entropy loss and is differentiable with respect to the model parameters $\theta$. The variance term $L_{\text{var}}$, seen in Equation 1, is differentiable except at the points where:

$$\sqrt{\mathbb{E}\left[(P - \mathbb{E}[P])^2\right]} = 0 \tag{7}$$

However, this is mitigated by the stability term $\epsilon$, where $\epsilon > 0$, to ensure that this condition is rarely met. The invariance term $L_{\text{inv}}$, seen in Equation 2, is differentiable as it is a quadratic function of $P$. Therefore, it is differentiable with respect to the model parameters $\theta$. The covariance term $L_{\text{cov}}$, seen in Equation 3, is differentiable for positive inputs, with the derivative $\frac{1}{2\sqrt{x}}$ for $x > 0$. However, this derivative approaches infinity as $x \to 0$, therefore, the gradient is not Lipschitz continuous near zero. For typical values encountered during training, it remains differentiable with respect to $\theta$.

We now establish the conditions under which FairVIC converges during optimisation. The gradients of the loss components are Lipschitz continuous *almost* everywhere and the loss function is bounded below by zero, we can guarantee converge to a critical point using gradient descent this is provided the learning rate $\eta$ satisfies the condition $0 < \eta < \frac{2}{L}$, where $L$ is the Lipschitz constant of the gradient of $L_{total}$. Under these conditions specifically, the following descent property holds:

$$L_{\text{total}}(\theta^{(t+1)}) \le L_{\text{total}}(\theta^{(t)}) - \eta\|\nabla_\theta L_{\text{total}}(\theta^{(t)})\|^2 + \frac{L\eta^2}{2}\|\nabla_\theta L_{\text{total}}(\theta^{(t)})\|^2 \qquad (8)$$

As $t \to \infty$, the norm of the gradient $L_{\text{total}}(\theta^{(t)})$ converges to zero, indicating convergence to a critical point.

## C  LAMBDA ABLATION STUDY RESULTS

Tables 7 and 8 show the full results for each model when the weights on the FairVIC terms are adapted. Table 7 shows the effect of changing $\lambda_{\text{acc}}$ while keeping the FairVIC terms equal where $\lambda_{\text{var, inv, cov}} = \frac{1-\lambda_{\text{acc}}}{3}$, and Table 8 sets $\lambda_{\text{acc}} = 0.1$, and suppresses two FairVIC terms to explore the effect of only utilising one term at a time. For full discussion and analysis of the results of the lambda ablation study, see Section 6.3.

Table 7: Performance and fairness results for FairVIC on the three tabular datasets, where the Fair-VIC terms are weighted equally, such that $\lambda_{\text{acc}} + \lambda_{\text{var}} + \lambda_{\text{inv}} + \lambda_{\text{cov}} = 1$.

| Dataset | $\lambda_{acc}$ | $\lambda_{var,inv,cov}$ | Accuracy | F1 Score | Equalized Odds | Absolute Odss | Demographic Parity | Disparate Impact |
|---|---|---|---|---|---|---|---|---|
| Adult Income | 0.10 | 0.30 | $0.8385 \pm 0.0044$ | $0.5957 \pm 0.0302$ | $0.2292 \pm 0.0524$ | $0.1255 \pm 0.0338$ | $-0.0368 \pm 0.0267$ | $0.8044 \pm 0.1546$ |
| | 0.20 | $0.2\overline{6}$ | $0.8428 \pm 0.0027$ | $0.6174 \pm 0.0144$ | $0.1889 \pm 0.0522$ | $0.0994 \pm 0.0276$ | $-0.0583 \pm 0.0166$ | $0.6959 \pm 0.0731$ |
| | 0.30 | $0.2\overline{3}$ | $0.8488 \pm 0.0031$ | $0.6374 \pm 0.0163$ | $0.0908 \pm 0.0437$ | $0.0564 \pm 0.0181$ | $-0.1042 \pm 0.0212$ | $0.5114 \pm 0.0603$ |
| | 0.40 | 0.20 | $0.8493 \pm 0.0034$ | $0.6408 \pm 0.0144$ | $0.0407 \pm 0.0100$ | $0.0328 \pm 0.0102$ | $-0.1284 \pm 0.0140$ | $0.4225 \pm 0.0348$ |
| | 0.50 | $0.1\overline{6}$ | $0.8501 \pm 0.0027$ | $0.6451 \pm 0.0122$ | $0.0485 \pm 0.0120$ | $0.0374 \pm 0.0146$ | $-0.1390 \pm 0.0150$ | $0.3926 \pm 0.0257$ |
| COMPAS | 0.10 | 0.30 | $0.6425 \pm 0.0131$ | $0.5848 \pm 0.0338$ | $0.0915 \pm 0.0386$ | $0.0698 \pm 0.0342$ | $-0.0226 \pm 0.0708$ | $0.9745 \pm 0.1178$ |
| | 0.20 | $0.2\overline{6}$ | $0.6557 \pm 0.0128$ | $0.6121 \pm 0.0235$ | $0.1145 \pm 0.0708$ | $0.0806 \pm 0.0465$ | $-0.1036 \pm 0.0624$ | $0.8471 \pm 0.0859$ |
| | 0.30 | $0.2\overline{3}$ | $0.6510 \pm 0.0129$ | $0.5993 \pm 0.0211$ | $0.1251 \pm 0.0497$ | $0.0870 \pm 0.0342$ | $-0.0873 \pm 0.0638$ | $0.8740 \pm 0.0849$ |
| | 0.40 | 0.20 | $0.6544 \pm 0.0152$ | $0.6059 \pm 0.0259$ | $0.1340 \pm 0.0568$ | $0.1019 \pm 0.0531$ | $-0.1167 \pm 0.0734$ | $0.8318 \pm 0.1030$ |
| | 0.50 | $0.1\overline{6}$ | $0.6643 \pm 0.0116$ | $0.6128 \pm 0.0225$ | $0.2071 \pm 0.0851$ | $0.1564 \pm 0.0766$ | $-0.1850 \pm 0.0772$ | $0.7459 \pm 0.0969$ |
| German Credit | 0.10 | 0.30 | $0.7145 \pm 0.0317$ | $0.8026 \pm 0.0275$ | $0.1123 \pm 0.0650$ | $0.0723 \pm 0.0414$ | $-0.0503 \pm 0.0510$ | $0.9385 \pm 0.0638$ |
| | 0.20 | $0.2\overline{6}$ | $0.7275 \pm 0.0238$ | $0.8132 \pm 0.0195$ | $0.1665 \pm 0.1037$ | $0.1196 \pm 0.0818$ | $-0.1088 \pm 0.1009$ | $0.8626 \pm 0.1238$ |
| | 0.30 | $0.2\overline{3}$ | $0.7385 \pm 0.0315$ | $0.8234 \pm 0.0238$ | $0.1832 \pm 0.0790$ | $0.1327 \pm 0.0514$ | $-0.0526 \pm 0.0877$ | $0.9369 \pm 0.1057$ |
| | 0.40 | 0.20 | $0.7330 \pm 0.0259$ | $0.8190 \pm 0.0202$ | $0.1085 \pm 0.0571$ | $0.0877 \pm 0.0502$ | $-0.1022 \pm 0.0752$ | $0.8706 \pm 0.0953$ |
| | 0.50 | $0.1\overline{6}$ | $0.7335 \pm 0.0319$ | $0.8188 \pm 0.0257$ | $0.1208 \pm 0.0589$ | $0.0931 \pm 0.0545$ | $-0.0833 \pm 0.0683$ | $0.8929 \pm 0.0851$ |

Table 8: Performance and fairness results for FairVIC on the three tabular datasets, with only one FairVIC term ($\lambda_{\text{var}}$, $\lambda_{\text{inv}}$, or $\lambda_{\text{cov}}$) weighted at a time.

| Dataset | $\lambda_{acc}$ | $\lambda_{var}$ | $\lambda_{inv}$ | $\lambda_{cov}$ | Accuracy | F1 Score | Equalized Odds | Absolute Odss | Demographic Parity | Disparate Impact |
|---|---|---|---|---|---|---|---|---|---|---|
| Adult Income | 0.10 | 0.90 | 0.00 | 0.00 | $0.8518 \pm 0.0032$ | $0.6620 \pm 0.0074$ | $0.0950 \pm 0.0261$ | $0.0808 \pm 0.0178$ | $-0.1761 \pm 0.0107$ | $0.3235 \pm 0.0335$ |
| | 0.10 | 0.00 | 0.90 | 0.00 | $0.8498 \pm 0.0039$ | $0.6506 \pm 0.0229$ | $0.0959 \pm 0.0314$ | $0.0820 \pm 0.0272$ | $-0.1715 \pm 0.0295$ | $0.3164 \pm 0.0274$ |
| | 0.10 | 0.00 | 0.00 | 0.90 | $0.8321 \pm 0.0070$ | $0.5640 \pm 0.0437$ | $0.2507 \pm 0.0758$ | $0.1382 \pm 0.0457$ | $-0.0256 \pm 0.0348$ | $0.8743 \pm 0.2136$ |
| COMPAS | 0.10 | 0.90 | 0.00 | 0.00 | $0.6499 \pm 0.0142$ | $0.5924 \pm 0.0321$ | $0.2634 \pm 0.0286$ | $0.2097 \pm 0.0238$ | $-0.2349 \pm 0.0226$ | $0.6872 \pm 0.0285$ |
| | 0.10 | 0.00 | 0.90 | 0.00 | $0.6598 \pm 0.0118$ | $0.6252 \pm 0.0212$ | $0.2858 \pm 0.0810$ | $0.2402 \pm 0.0794$ | $-0.2776 \pm 0.0805$ | $0.6202 \pm 0.0969$ |
| | 0.10 | 0.00 | 0.00 | 0.90 | $0.6504 \pm 0.0169$ | $0.6182 \pm 0.0258$ | $0.0984 \pm 0.0565$ | $0.0694 \pm 0.0486$ | $0.0007 \pm 0.0718$ | $1.0155 \pm 0.1383$ |
| German Credit | 0.10 | 0.90 | 0.00 | 0.00 | $0.7385 \pm 0.0316$ | $0.8207 \pm 0.0269$ | $0.1597 \pm 0.0614$ | $0.1330 \pm 0.0497$ | $-0.1386 \pm 0.0724$ | $0.8209 \pm 0.0951$ |
| | 0.10 | 0.00 | 0.90 | 0.00 | $0.7125 \pm 0.0238$ | $0.8003 \pm 0.0203$ | $0.1665 \pm 0.0977$ | $0.1268 \pm 0.0682$ | $-0.1191 \pm 0.0864$ | $0.8450 \pm 0.1129$ |
| | 0.10 | 0.00 | 0.00 | 0.90 | $0.7155 \pm 0.0374$ | $0.8023 \pm 0.0321$ | $0.2321 \pm 0.1504$ | $0.1565 \pm 0.1000$ | $0.0945 \pm 0.0979$ | $1.1274 \pm 0.1427$ |

