# OpenReview forum: "Learning Fairer Representations with FairVIC"
_ICLR.cc/2025/Conference — ICLR 2025 Conference Withdrawn Submission_

### Official Review · Reviewer_UFMC · 2024-11-01

**Soundness:** 1
**Presentation:** 3
**Contribution:** 1
**Rating:** 3
**Confidence:** 4

**Summary:**

This paper introduces three loss terms designed to regulate the learning process and enhance fairness, without specifying the fairness metrics to be optimized during the training stage. The paper conducts experiments on both tabular data and one NLP dataset to demonstrate its effectiveness.

**Strengths:**

This method is straightforward and comprehensible.

The writing style is clear and easy to follow.

Figure 1 effectively illustrates the distinctions among various methods.

**Weaknesses:**

**Technical flaw:** My primary concern pertains to this point. The author introduces three loss terms in equations (1), (2), and (3). However, terms (1) and (2) *have no direct dependence on the network parameters*. Therefore, the gradient of the parameters with respect to those losses is always zero, making optimization of these terms nonsensical. This constitutes a significant technical flaw.

**Lack of evaluation:**  Given the plethora of in-processing methods available in fairness and robustness research, the criteria for selecting the baseline model remain unclear. I recommend the inclusion of a representative method, such as GroupDRO, for comparison, along with Min-max fairness for a more comprehensive evaluation.

**Experiment flaw:** How can Demographic Parity attain a negative value? (line 157)

**Questions:**

Please refer to the identified weaknesses; I have no further questions regarding this manuscript.

---

### Official Review · Reviewer_8uT6 · 2024-11-03

**Soundness:** 2
**Presentation:** 2
**Contribution:** 1
**Rating:** 3
**Confidence:** 5

**Summary:**

The authors introduce an in-processing method that attempts to control how the predictions vary w.r.t. sensitive attributes. The proposed approach introduces three regularizers aimed at encouraging both group and individual notions of fairness simultaneously. They show how the proposed approach compares favorably with other in-processing approaches on tabular data.

**Strengths:**

The authors propose fairness regularizers that aim to satisfy notions of group, individual, and counterfactual fairness simultaneously

For the chosen datasets, the method is evaluated extensively against the chosen baselines. While there are a few additions and ablations I hope to see, the results are already considerable. I like the visualization provided by Figure 1.

**Weaknesses:**

The introduction of the regularizers is somewhat instructable. Basic questions like how do they regularizers depend on the model’s predictions are left unanswered.

The dismissal of prior approaches (e.g. regularizing using a metric grounded in legal precedent, post-processing methods at large) was rather uncharitable, in my opinion. What the authors do is describe potential limitations of these methods, but I think the claims are not properly substantiated, and a more detailed account for how the proposed method contrasts with these approaches would help.

The method is quite complex, and (despite the many experiments and ablations) I wasn’t convinced that all components are necessary. To handle this complexity, the authors propose a way to adaptively choose hyperparameters, but it seems flawed to me.

**Questions:**

Additional questions/comments:
* [L015-019] Seems to me like grounding the ("user-defined") fairness definitions in normative/philosophical/legal arguments for when differential treatment is acceptable [https://proceedings.mlr.press/v81/binns18a.html, https://proceedings.mlr.press/v81/binns18a.html] could actually be a strength of current fair ML approaches. I understand that this implies ambiguity over which metric is appropriate, but from what I see in the FAccT space practitioners seem to be relatively comfortable with this, so long as domain experts can be found to provide guidance on the choice of metric. To put it another way, the authors claim such approaches are “ad-hoc” [L051]; I would agree they are ad-hoc if implemented carelessly, but can be intentional and interpretable when integrated into a thoughtful design process that includes non-technologists.
* [L053] The cited Berk paper seems like an odd choice to substantiate the claim about the issues with post-processing methods; to my knowledge those authors don't compare their in-processing method against post-processing baselines. Setting the citation aside, if you care about the final predictions/decisions then a good post-processing method should be fine in principle, right? I sort of understand the appeal to go beyond post-processing and "[address] the underlying biases in the data and model", but I think that argument makes the most sense if you approach the problem from a representation learning angle, where a single model must accomplish multiple downstream tasks or be transferred to some new setting, which doesn't seem to match the experimental setting considered here. If the authors really believe this argument then they should compare against a post-processing method empirically.
* Since the proposed approach uses concepts like variance and invariance to encode fairness, I feel that prior work attempting to regularize fair classifiers/representations using information theory-inspired priors should be acknowledged [https://proceedings.neurips.cc/paper/2018/hash/415185ea244ea2b2bedeb0449b926802-Abstract.html, https://proceedings.mlr.press/v89/song19a, https://proceedings.mlr.press/v97/creager19a.html].
* [L058] Why are the concepts of variance, invariance and covariance "more principles" than alternative?
* [L196, L205] Is P not a random variable which is a component of the dataset? How are Eqn 1 and Eqn 2 dependent on the model? For Eqn 3 the dependence (through $\hat y$) is clear.
* The variance and invariance terms seem structurally quite similar. Are the authors sure both are needed? If yes, please provide an intuition as to why. I appreciate that many configurations of lambdas (including zero values) are tried in the appendix, but I wasn't able to find a straightforward ablation where, for each regularizer R in {variance, invariance, covariance}, drop R from the loss, tune other hyperparameters the best you can, and report FairVIC-minus-R results for all datasets, ideally in an easy-to-interpret format (like Fig 1).
* [Sec 4.2] Tuning hyperparameters from training data seems like a dangerous game to play. Also now you need to introduce a hyperprior $R$ (I couldn't find discussion of how this is selected), which means the practitioner needs an intuitive understanding of what scale is appropriate for lambda. If that is the case, why not just do a standard hyperparameter search over a reasonable range (which must be known, otherwise $R$ is misspecified) using validation data?
* [L269] How is the regularization term $R$ implemented?
* [Table 1] These tables contain a lot of information. Formatting the best results (respecting the uncertainty estimates from the std devs) in bold would help readability. I do think the visualization in Figure 1 also helps interpret the results, for what it's worth.
* [Table 2] If the lambdas are tuned adaptively, why are the final results such clean numbers? Was some sort of quantization applied?
* [L1002] the authors show the effect of optimizing with just one term at a time. I hesitate to ask this because the experiments are already rather broad, but I was curious what would happen if only one term was *omitted* at a time.

---

### Official Review · Reviewer_CVTG · 2024-11-04

**Soundness:** 2
**Presentation:** 3
**Contribution:** 2
**Rating:** 5
**Confidence:** 3

**Summary:**

The paper introduces a novel approach called FairVIC for learning fair representations to mitigate bias in classification models. Inspired by Variance-Invariance-Covariance Regularization from self-supervised learning, FairVIC incorporates a regularizer with three key terms: the variance term, which reduces reliance on the protected attribute as a trivial solution; the invariance term, which ensures consistent predictions for similar individuals; and the covariance term, which minimizes dependence on the protected attribute. Consequently, the FairVIC loss function optimizes for both group and individual fairness simultaneously. Additionally, the paper presents an adaptive multi-objective gradient descent method to dynamically adjust the penalty weights. The authors demonstrate FairVIC's effectiveness by empirically evaluating it on three tabular classification problems and one text classification problem, comparing it against five baseline methods.

**Strengths:**

1. This paper introduces a novel fairness regularizer that simultaneously optimizes for both group and individual fairness.

2. The proposed method is evaluated across two modalities, demonstrating optimal performance in terms of the fairness-accuracy trade-off when compared to baseline methods.

3. Additionally, the paper presents a method for automatically tuning the weights of the regularizer.

**Weaknesses:**

1. The evaluation is limited by the number of datasets used. It would be beneficial to test the proposed method on a wider range of tabular and text classification datasets [1, 2, 3].

2. FairVIC shares similarities with both fair representation learning methods and fairness penalty approaches; however, the authors only compare it with the adversarial debiasing algorithm by Zhang et al. It would be helpful to compare FairVIC with other fairness regularizers, such as those addressing equalized odds [4], disparate impact [5], and individual fairness.

3. The presentation of the tables could be improved by highlighting the best-performing algorithm for each dataset and calculating the average rank of each method.


[1] Ding, F., Hardt, M., Miller, J. and Schmidt, L., 2021. Retiring adult: New datasets for fair machine learning. Advances in neural information processing systems, 34, pp.6478-6490.

[2] De-Arteaga, M., Romanov, A., Wallach, H., Chayes, J., Borgs, C., Chouldechova, A., Geyik, S., Kenthapadi, K. and Kalai, A.T., 2019, January. Bias in bios: A case study of semantic representation bias in a high-stakes setting. In proceedings of the Conference on Fairness, Accountability, and Transparency (pp. 120-128).

[3] https://www.kaggle.com/c/jigsaw-unintended-bias-in-toxicity-classification

[4] Hardt, M., Price, E. and Srebro, N., 2016. Equality of opportunity in supervised learning. Advances in neural information processing systems, 29.

[5] Zafar, M.B., Valera, I., Rogriguez, M.G. and Gummadi, K.P., 2017, April. Fairness constraints: Mechanisms for fair classification. In Artificial intelligence and statistics (pp. 962-970). PMLR.

[6] Dwork, C., Hardt, M., Pitassi, T., Reingold, O. and Zemel, R., 2012, January. Fairness through awareness. In Proceedings of the 3rd innovations in theoretical computer science conference (pp. 214-226).

**Questions:**

1. Could the authors please clarify the differences between the variance and covariance terms in terms of their impact on the model? Both seem to reduce the model's reliance on the protected attribute, but a more detailed explanation would be helpful.

2. The regularization terms in FairVIC appear to optimize demographic parity by reducing the model's dependence on the sensitive attribute, as well as individual fairness. How does the model also learn to optimize for the Equalized Odds metric?

3. I am particularly interested in seeing if FairVIC outperforms a model that combines demographic parity and equalized odds fairness regularizers added to the loss objective.

---

### Official Review · Reviewer_prYg · 2024-11-06

**Soundness:** 2
**Presentation:** 3
**Contribution:** 2
**Rating:** 5
**Confidence:** 4

**Summary:**

The paper proposes FairVIC, a method to learn fair representations by adding terms for variance invariance and covariance, to remove any correlation between sensitive attributes and outcomes. FairVIC borrows principles from self supervised learning and applies it to reduce bias in outcomes of a neural network. FairVIC adds 3 new terms (variance, invariance, and covariance) to the traditional empirical risk minimization objective which can be balanced dynamically. Empirical results show that FairVIC is effective at reducing a variety of biases all at once, while also being more robust than existing methods.

**Strengths:**

- The method is broadly applicable to reduce a variety of biases at the same time
- Weights on new loss terms can be dynamically balanced
- The additional loss terms do not introduce a large computational overhead on the forward or backward passes

**Weaknesses:**

See questions

**Questions:**

- [Fairness Through Unawareness] This sounds like fairness behind a "veil of ignorance" where the claim is that removing sensitive attribute will prevent all kinds of downstream unfairness. However, for a long time it has been known in the fairness community that this approach is only one of the possible definitions [1] and in some cases might not be enough since there can be correlated non-sensitive attributes due to historical discrimination (eg: zipcodes can predict race in large part due to redlining laws in mid 1900s) [2].

 - [Missing comparison to constrained fairness methods] Especially on tabular datasets considered in the paper, there have been many prior works (eg [5]) that use traditional ML models to achieve a much better fairness / accuracy tradeoff. I would have liked to see some comparison to such works.

 - [Missing discussion of conflicts of fairness definitions] Given many well studied cases of inherent conflicts in fairness definitions [3], I found the claims of achieving fairness along multiple definitions simultaneously a bit concerning. Generally there's a lack of discussion in the paper about possible conflicts in these definitions and when such conflicts happen which definition will prevail under FairVIC.

 - [DI and DP are the same definition] The paper presents Disparate Impact and Demographic Parity as different definitions but these are the same definition of fairness (see [5]). Thus, I'm also not super sure how to interpret Table 1 and Figure 1 results since some methods show reduction in DI but not in DP and vice versa.

 - [More datasets where deep learning models shine] Since the proposed method should work especially well on deep neural nets, I would have liked to see some datasets where deep learning models are really necessary (traditional ML models perform very well already on the considered tabular datasets). Some examples would be CheXpert and CelebA which have been used in prior fair deep learning literature.

[1] https://arxiv.org/abs/1908.09635

[2] https://arxiv.org/abs/1808.00023

[3] https://arxiv.org/abs/1609.05807

[4] https://fairmlbook.org

[5] https://www.jmlr.org/papers/v20/18-262.html

---

### Note · Authors · 2024-11-23

**Comment:**

We greatly appreciate the reviewers’ thoughtful and detailed feedback. Their insights have highlighted areas where the paper can be refined and strengthened, and we will carefully consider these points as we work to improve the paper in future iterations.

**Withdrawal Confirmation:**

I have read and agree with the venue's withdrawal policy on behalf of myself and my co-authors.